# Brief communication: Chest radiography score in young COVID-19 patients: Does one size fit all?

**Gioele Castelli**[1], **Umberto Semenzato**[1], **Sara Lococo**[1], **Elisabetta Cocconcelli**[1], **Nicol Bernardinello**[1], **Giulia Fichera**[2], **Chiara Giraudo**[2], **Paolo Spagnolo**[1], **Annamaria Cattelan**[3], **Elisabetta Balestro**[1]*

**1** Respiratory Disease Unit, Department of Cardiac, Thoracic, Vascular Sciences and Public Health, University of Padova and Padova City Hospital, Padova, Italy, **2** Institute of Radiology, Department of Medicine, University of Padova, Padova, Italy, **3** Division of Infectious and Tropical Diseases, Azienda Ospedaliera and University of Padova, Padova, Italy

* elisabetta.balestro@aopd.veneto.it

## Abstract

During the SARS-CoV-2 pandemic, chest X-Ray (CXR) scores are essential to rapidly assess patients' prognoses. This study evaluates a published CXR score in a different national healthcare system. In our study, this CXR score maintains a prognostic role in predicting length of hospital stay, but not disease severity. However, our results show that the predictive role of CXR score could be influenced by socioeconomic status and healthcare system.

## Introduction

Since the beginning of the current pandemic, severe acute respiratory syndrome coronavirus 2 (SARS-CoV-2) has already affected over 110,000,000 people causing more than 2,400,000 deaths. Portable chest radiography (CXR) demonstrated to be a useful tool to diagnose and monitor the disease in emergency departments, intensive care units (ICU), and Coronavirus Disease-19 (COVID-19) wards [1]. CXR has been used as a prognostic tool in acute respiratory distress syndrome even before COVID-19 [2]. From the beginning of the pandemic, several CXR scores have been proposed, especially to triage patients within emergency departments [3–6]. These different scores, such as the CARE score or the BRIXIA score, correlated to clinical conditions at admission and to clinical outcomes [7–9]. Among these, the one of Toussie et al. demonstrated a good prognostic value in patients between the ages of 21 and 50 years with COVID-19 [10]. Our study aimed to assess the value of this score on young and middle-aged Italian COVID-19 patients.

## Materials and methods

In this retrospective study, approved by our local ethic committee (Comitato Etico per la Sperimentazione Clinica della Provincia di Padova, protocol n˚46430/03.08.2020, which waived

**Data Availability Statement:** The minimal data files are available from the OSF database (https://osf.io/y97mh/?view_only=216b7558d9114e2eb2b7a577edcc85eb).

**Funding:** The author(s) received no specific funding for this work.

**Competing interests:** The authors have declared that no competing interests exist.

the need for patient's informed consent), we evaluated the role of the CXR score proposed by Toussie et al. in a cohort of 51 young and middle-aged patients consecutively hospitalized for SARS-CoV-2 infection in the University Hospital of Padua from February to August 2020. Patients were included if underwent either digital anteroposterior CXR or digital posteroanterior and lateral chest radiography, all performed in a single Radiology Unit of our Hospital dedicated to COVID-19 patients from the Emergency Department, with the same CXR equipment and setting. If patients accessed the Emergency Department more than once, the CXR of the access leading to hospitalization was assessed. CXR was divided into 6 zones, using the upper and lower hilar marking as limits. Each zone with an opacity was counted as a point, with the score ranging from 0 to 6 points. For the whole population, two radiologists (10 and 4 years experience) scored the initial CXR independently of each other. To minimize bias, reviewers were blinded to patient histories other than COVID-19 positivity. We considered a CXR score equal or major to 3 as a cut-off to categorize patients as follows: low and high radiological risk scores (LRRS and HRRS, respectively) [10]. Clinical outcomes assessed in our cohort were: the need for high-intensity medical setting (ICU or Respiratory ICU), need for invasive ventilation, prolonged hospitalization (>10 days). The Cohen's kappa coefficient was used to assess agreement in CXR interpretation between the two radiologists. The fatality rate was reported. Mann-Whitney, $\chi2$ and Fisher's exact test were performed for comparison, as appropriate. The univariate logistic regression analysis was applied to evaluate if age, Body Mass Index (BMI), smoking history, ethnicity, length of symptoms, hypertension, asthma, diabetes, and CXR score influenced clinical outcomes. Statistics were performed using SPSS (v26, IBM Armonk, NY, USA) (level of significance $p<0.05$).

## Results

Patients characteristics and categorization are summarized in the Table 1, examples of CXR score are in Fig 1. All patients accessed the Emergency Department only once before the hospiatlization. Patient underwent either digital anteroposterior CXR (47 of 51, 92%) or digital posteroanterior and lateral chest radiography (4 of 51, 8%). CXRs were scored by two radiologists with a moderate to substantial agreement with Choen's kappa coefficient ranging from 0.55 to 0.66 in different lobes.

We observed a higher percentage of patients with LRRS (n = 41; 80%) compared to Toussie's (n = 87; 60%). Subjects with HRRS were significantly older than the LRRS (p = 0.04), they presented a lower arterial oxygen partial pressure to fractional inspired oxygen. (P/F ratio) (p = 0.02), and had a higher BMI (p = 0.03). Among the patients with normal BMI (i.e., <25), only one had HRRS.

At univariate analyses, HRRS was a risk factor for prolonged hospitalization (OR 4.65, CI95%:1.09–19.87;p = 0.03) whereas HRRS did not influence the need of high-intensity medical setting and intubation. The multivariate analysis revealed that HRRS (OR 7.83, CI95%:1.23–49.99;p = 0.03), diabetes (21.72,1.63–289.40;p = 0.02), and the need of high-intensity medical setting (51.30,4.53–582.71;p = 0.001), were independent risk factors for prolonged hospitalization.

## Discussion

In young and middle-aged COVID-19 patients we observed a higher percentage of LRRS than Toussie and colleagues (80% vs 60%). This difference could be due to the different healthcare systems. In fact, in countries with public healthcare, patients with mild symptoms are presumably more prone to access health services [11]. Confirming previous data, patients presenting HRRS had worst pulmonary gas exchanges, with a lower P/F ratio [1, 5, 12]. Moreover, HRRS

**Table 1. Baseline demographics and clinical features of the overall young population hospitalized for SARS-CoV-2 related infection, and of the two subgroups categorized in *low (LRRS, 0–2)* and *high (HRRS, 3–6)* radiological risk score.**

| | Overall Population (n = 51) | Low radiological risk score (LRRS) (n = 41) | High radiological risk score (HRRS) (n = 10) | p Value |
|---|---|---|---|---|
| **Male–*n (%)*** | 30 (59) | 24 (59) | 6 (60) | 0.33 |
| **Age at admission–*years*** | 43 (34–48) | 41 (32–48) | 45.5 (43–49) | **0.04** |
| **Race (ethnicity)–*n (%)*** | | | | |
| • **Caucasian** | 37 (72) | 28 (68) | 9 (90) | 0.72 |
| • **Asian** | 4 (8) | 4 (10) | 0 (0) | |
| • **Black** | 8 (16) | 7 (17) | 1 (10) | |
| • **Other** | 2 (4) | 2 (5) | 0 (0) | |
| **Smoking history–*pack years*** | 0 (0–0) | 0 (0–1) | 0 (0–0) | 0.37 |
| • **Current–*n (%)*** | 6 (12) | 5 (12) | 1 (10) | 0.39 |
| • **Former–*n (%)*** | 7 (14) | 7 (17) | 0 (0) | |
| • **Nonsmokers–*n (%)*** | 38 (74) | 29 (71) | 9 (90) | |
| **BMI (kg/m^2)** | 26 (22.6–29.5) | 24.7 (22.2–27.9) | 28.6 (27–30) | **0.03** |
| **BMI ≥ 25 –*n (%)*** | 28 (55) | 19 (46) | 9 (90) | **0.03** |
| **Time from symptoms onset to CXR–*days*** | 5 (2–7) | 4 (2–6) | 5.5 (3–8) | 0.40 |
| **Hospitalization—*days*** | 7 (3–12) | 6 (3–10) | 13 (7–23) | **0.005** |
| **Comorbidities—*n (%)*** | | | | |
| • **Asthma** | 3 (6) | 2 (5) | 1 (10) | 0.48 |
| • **Hypertension** | 9 (18) | 8 (20) | 1 (10) | 0.66 |
| • **Diabetes type II** | 5 (10) | 3 (7) | 2 (20) | 0.25 |
| • **HIV** | 0 (0) | 0 (0) | 0 (0) | 1.00 |
| **Febrile[†] at ED presentation–*n(%)*** | 43 (84) | 33 (80) | 10 (100) | 0.33 |
| **P/F ratio** | 398 (316–425) | 400 (336–429) | 302 (188–391) | **0.02** |
| **High-intensity medical care[‡] –*n(%)*** | 8 (16) | 6 (15) | 2 (20) | 0.50 |
| **Invasive ventilation–*n(%)*** | 3 (6) | 2 (5) | 1 (10) | 0.48 |
| **Dead–*n (%)*** | 1 (2) | 0 (0) | 1 (10) | 0.19 |

Values are expressed as numbers and (%) or median and Q1 –Q3, as appropriate. To compare demographic between LRRS and HRRS, χ2 and Fisher's exact test (n < 5) for categorical variables and Mann-Whitney test for continuous variables were used. p values in bold (< .05) show significance.

BMI = body mass index; HIV = human immunodeficiency virus; CXR = chest radiography; P/F.ratio = arterial oxygen partial pressure to fractional inspired oxygen.

[†] febrile is defined by temperature over 38˚C.

[‡]high-intensity medical care is defined as the necessity of at least one between invasive/non-invasive ventilation or high-flow nasal cannula.

subjects showed higher BMI values compared to LRRS. Our result suggests that being overweight, and not necessarily obese, represents a risk factor for HRRS and leads to a worse prognosis. This finding is in line with the results of a larger multicentre study of young and middle-aged Italian patients, where obesity and older age were independent predictors for mechanical ventilation [12]. Some authors suggest that fatality and severity of SARS-CoV-2 infection are higher in lower socioeconomic classes [13]. Given the known association of lower socioeconomic status and higher BMI, a stratification risk model including such information should be explored [14].

In our population, HRRS was an independent prognostic factor for prolonged hospitalization, but not for intubation. This concept is of interest if we consider that in our study the interval between the onset of symptoms and CXR is similar between the two groups, opposed to the Toussie's study. Limitation of our study are the small sample size, the monocentric and restrospective nature of study design. In conclusion, our preliminary results confirm that CXR

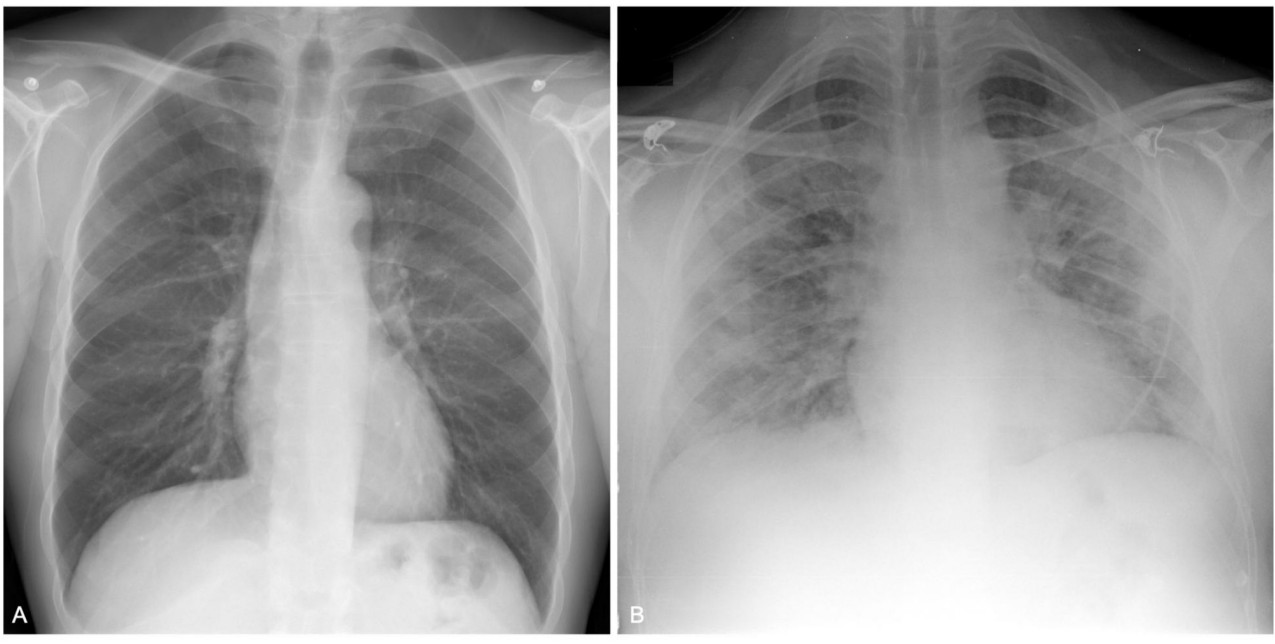

**Fig 1. Examples of a chest severity score.** A. Chest radiograph of a 48-year-old male. CXR does not show any opacities; total score = 0. B. CXR of a 42-year-old male shows opacities in all three right lung zones and in the left middle and lower lung zones; total score = 5.

scores are useful for the management of COVID-19 patients but also point out that their prognostic role might be influenced by the socioeconomic background and the type of healthcare system. CXR scores should be integrated in a multiparametric score system including patients' characteristics and clinical findings.

## Author Contributions

**Conceptualization:** Gioele Castelli, Umberto Semenzato, Sara Lococo, Elisabetta Balestro.

**Data curation:** Gioele Castelli, Umberto Semenzato, Sara Lococo, Nicol Bernardinello.

**Formal analysis:** Gioele Castelli, Umberto Semenzato, Sara Lococo, Elisabetta Balestro.

**Investigation:** Gioele Castelli, Umberto Semenzato, Sara Lococo, Nicol Bernardinello.

**Supervision:** Elisabetta Cocconcelli, Paolo Spagnolo, Annamaria Cattelan, Elisabetta Balestro.

**Validation:** Giulia Fichera, Chiara Giraudo.

**Writing – original draft:** Gioele Castelli, Umberto Semenzato, Sara Lococo, Elisabetta Balestro.

**Writing – review & editing:** Elisabetta Cocconcelli, Paolo Spagnolo, Annamaria Cattelan, Elisabetta Balestro.

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
