## [Decision Letter · Decision Letter 0]

14 Jul 2021

PONE-D-21-07724

Brief communication: chest radiography score in young COVID-19 patients: does one size fit all?

PLOS ONE

Dear Dr. Balestro,

Thank you for submitting your manuscript to PLOS ONE. After careful consideration, we feel that it has merit but does not fully meet PLOS ONE’s publication criteria as it currently stands. Therefore, we invite you to submit a revised version of the manuscript that addresses the points raised during the review process.

We look forward to receiving your revised manuscript.

Kind regards,

Eman Sobh, M.D.

Academic Editor

PLOS ONE

Journal Requirements:

2. Please amend your Methods section to state the full name of your institutional ethics board and please provide the same information stated in your Ethics Statement regarding the waiver of consent.

Reviewers' comments:

Reviewer's Responses to Questions

**Comments to the Author**

1. Is the manuscript technically sound, and do the data support the conclusions?

Reviewer #1: No

Reviewer #2: Partly

Reviewer #3: Yes

2. Has the statistical analysis been performed appropriately and rigorously? 

Reviewer #1: No

Reviewer #2: Yes

Reviewer #3: Yes

3. Have the authors made all data underlying the findings in their manuscript fully available?

Reviewer #1: No

Reviewer #2: Yes

Reviewer #3: Yes

4. Is the manuscript presented in an intelligible fashion and written in standard English?

Reviewer #1: Yes

Reviewer #2: Yes

Reviewer #3: Yes

5. Review Comments to the Author

Reviewer #1: The study design as mentioned lack the good number of Patients but the most important clinical error it lacks standardization as we know CXR to reveal opacities and lung shadows specially consolidation and GGO in cases of COVID 19 it should be standard. Exposure is one factor, type of CXR PA versus AP will affect the reading of CXR , we know also the different radiologists may read Same CXR differently and inter observer variability if a factor that limit CXR scoring , these points were not clearly delineated in the study

There is overlap between different causes of GGO and consolidations even in COVID as pulmonary edema and pleural effusion that might be misinterpreted in CXR as consolidation, it was not clearly identified in the study

Reviewer #2: The manuscript must be checked by any grammar program.

Abstract

• Amend “patients’ prognosis” To “patients’ prognoses”

• Amend “In our study” To “In our study,”

Introduction

The introduction was too short and I think it will be better to conclude more studies about CXR scores and its prognostic value.

• Amend “Aim of our study was to assess the value of this” To “Our study aimed to assess the value of this”

Materials and Methods

1. Description of the methodology was concise and contained summarized description of well-established selection parameters.

2. The number of patients was suitable (51 patients) but it could be statistically accepted.

3. The choice of young and middle age was interesting.

4. Inclusion and Exclusion criteria seem logical.

5. Amend “CXR was divided in” To “CXR was divided into”

6. Amend “score equal or major to 3 as cut-off to categorize” To “score equal or major to 3 as a cut-off to categorize”

7. Amend “Clinical outcomes assessed in our cohort were: need for high-intensity” To “Clinical outcomes assessed in our cohort were: the need for high-intensity”

Results and Discussion

1. The results were represented by nice and informative radiographic images.

2. The study findings were adequately discussed.

Research criteria (1 = Excellent) (2 = Good) (3 = Fair) (4 = poor)

Originality 3

Contribution to the field 3

Technical quality 3

Clarity of Presentation 2

Depth of Research 3

Decision

The study is interesting. Therefore, I recommend it for publication after minor corrections.

Reviewer #3: The present brief communication provides interesting data on role chest radiography score in young COVID-19 patients. The study is well presented and proper analysed. Just a few suggestions:

1) it is known from literature that higher d-dimer level is significantly associated with a worse prognosis even in young and middle aged patients (i-e. Bonifazi et al Journal of Clinical Medicine 2021). It would be interesting, if available, to include baseline values in the multivariate analysis. If not available, i suggest to mention the lack of infomation on clinical features likely to influence outcomes, as limitation.

2) Were baseline CTs available for study population? if yes, il would be interesting to correlate CT scores with Rx scores

3) Among study limitations i would mention also the retrospective nature of the study

6. PLOS authors have the option to publish the peer review history of their article (what does this mean?). If published, this will include your full peer review and any attached files.

Reviewer #1: **Yes: **Prof Hassan Mohamed Ahmed Aref Shabana

Reviewer #2: **Yes: **Amr A. Abd-Elghany

Reviewer #3: No

---

## [Author Response · Author response to Decision Letter 0]

2 Aug 2021

Journal Requirements: 

The manuscript has been amended as requested.

2. Please amend your Methods section to state the full name of your institutional ethics board and please provide the same information stated in your Ethics Statement regarding the waiver of consent.

The full name of the institutional ethics board and the waiver of consent has been added (page 3 lines 43-45)

Reviewers' comments:

Reviewer #1, comment 1: The study design as mentioned lack the good number of Patients but the most important clinical error it lacks standardization as we know CXR to reveal opacities and lung shadows specially consolidation and GGO in cases of COVID 19 it should be standard. 

Response to reviewer #1, comment 1: As stated also by the reviewer we mentioned that sample size is a limitation of our contribution. However, it is important to underline that all subjects were fully characterized, and consecutively collected in order to reduce selection bias. Moreover, regarding CXR standardization it’s crucial to mention the peculiar health system organization from the beginning of pandemic with a single Radiology Unit dedicated to COVID-19 patients from the Emergency Department, with the same CXR equipment, staff and setting. This point is now better clarified in the Material and Methods section (page 3, lines 49-51). Furthermore, multiple CXR scores have already been standardized as prognostic tools both in COVID-19 and ARDS, as we now included in the introduction (page 3, lines 34-38).

Reviewer #1, comment 2: Exposure is one factor, type of CXR PA versus AP will affect the reading of CXR , we know also the different radiologists may read Same CXR differently and inter observer variability if a factor that limit CXR scoring , these points were not clearly delineated in the study 

Response to reviewer #1, comment 2: we agree with reviewer’s comment and according to Toussie’s score method, which we aimed to validate in a different cohort, patients were included if underwent either digital anteroposterior chest radiography (47 of 51, 92%) or digital posteroanterior and lateral chest radiography (4 of 51, 8%). This point is now better explained in the Materials and Methods section (page3, lines 48, 49) and Results section. Interobserver variability was a major concern, thus readers were trained for applying Toussie’s CXR score, obtaining a moderate to substantial interobserver concordance among two different readers (Cohen's kappa coefficient (κ) ranging from 0.55 to 0.66 in different lobes).

Reviewer #1, comment 3: There is overlap between different causes of GGO and consolidations even in COVID as pulmonary edema and pleural effusion that might be misinterpreted in CXR as consolidation, it was not clearly identified in the study.

Response to reviewer #1, comment 3: We agree with the reviewer, CXR interpretation can be confounded by comorbid conditions, like heart failure or chronic lung disease. However, considering that our patients were under 50 years old, the incidence of comorbidities that could lead to CXR misinterpretation within this age was very low (3 of 51, 6%). 

Reviewer #2: The manuscript must be checked by any grammar program.

Abstract

• Amend “patients’ prognosis” To “patients’ prognoses”

• Amend “In our study” To “In our study,”

Introduction

The introduction was too short and I think it will be better to conclude more studies about CXR scores and its prognostic value.

• Amend “Aim of our study was to assess the value of this” To “Our study aimed to assess the value of this”

Materials and Methods

1. Description of the methodology was concise and contained summarized description of well-established selection parameters.

2. The number of patients was suitable (51 patients) but it could be statistically accepted.

3. The choice of young and middle age was interesting.

4. Inclusion and Exclusion criteria seem logical.

5. Amend “CXR was divided in” To “CXR was divided into”

6. Amend “score equal or major to 3 as cut-off to categorize” To “score equal or major to 3 as a cut-off to categorize”

7. Amend “Clinical outcomes assessed in our cohort were: need for high-intensity” To “Clinical outcomes assessed in our cohort were: the need for high-intensity”

Results and Discussion

1. The results were represented by nice and informative radiographic images.

2. The study findings were adequately discussed.

Research criteria (1 = Excellent) (2 = Good) (3 = Fair) (4 = poor)

Originality 3

Contribution to the field 3

Technical quality 3

Clarity of Presentation 2

Depth of Research 3

Decision

The study is interesting. Therefore, I recommend it for publication after minor corrections.

Response to reviewer #2. We would like to thank the reviewer for all the comments. We amended the manuscript accordingly to your suggestions as you can see in the revised version. Moreover, in the introduction, we now mentioned a larger number of studies on CXR scores and their prognostic value (page 3 lines 34-38). We are glad that the reviewer had considered our results well discussed and our methodology suitable for this study. We are aware of the small sample size (as we stated in study limitations), however the limited number was also due to the consecutive enrollment we on purpose applied to reduce selection bias.

Reviewer #3: The present brief communication provides interesting data on role chest radiography score in young COVID-19 patients. The study is well presented and proper analysed. Just a few suggestions:

1) it is known from literature that higher d-dimer level is significantly associated with a worse prognosis even in young and middle aged patients (i-e. Bonifazi et al Journal of Clinical Medicine 2021). It would be interesting, if available, to include baseline values in the multivariate analysis. If not available, i suggest to mention the lack of infomation on clinical features likely to influence outcomes, as limitation.

2) Were baseline CTs available for study population? if yes, il would be interesting to correlate CT scores with Rx scores

3) Among study limitations i would mention also the retrospective nature of the study

Response to reviewer #3. We thank the reviewer for comments and suggestions that improved our manuscript.

1) The cited article is very interesting and supports our results on the importance of the BMI in young COVID-19 patients, we did not quoted it before because it was not published at the time of our submission; now it is mentioned in page 6 lines 94-96. We then repeated the multivariate analysis including baseline D-dimer level, that was available in our subjects. The new analysis confirmed that high radiological risk score, diabetes and the need of high-intensity medical setting were independent risk factors for prolonged hospitalization also including baseline D-dimer, that was not a risk factor itself (OR 0.998. CI 95%: 0.990-1.005; p=0.556). 

2) Unfortunately baseline CTs in this cohort were not available, because in our hospital during the first wave, CT scans were not largely and easily available for all patients, as it would have been later in the second wave. 

3) Thank you for the suggestion, we included the suggested limitation in the discussion (page 7, line 103).

---

## [Decision Letter · Decision Letter 1]

20 Oct 2021

PONE-D-21-07724R1Brief communication: chest radiography score in young COVID-19 patients: does one size fit all?PLOS ONE

Dear Dr. Balestro,

Thank you for submitting your manuscript to PLOS ONE. After careful consideration, we feel that it has merit but does not fully meet PLOS ONE’s publication criteria as it currently stands. Therefore, we invite you to submit a revised version of the manuscript that addresses the points raised during the review process.

We look forward to receiving your revised manuscript.

Kind regards,

Eman Sobh, M.D.

Academic Editor

PLOS ONE

Reviewers' comments:

Reviewer's Responses to Questions

**Comments to the Author**

1. If the authors have adequately addressed your comments raised in a previous round of review and you feel that this manuscript is now acceptable for publication, you may indicate that here to bypass the “Comments to the Author” section, enter your conflict of interest statement in the “Confidential to Editor” section, and submit your "Accept" recommendation.

Reviewer #1: (No Response)

Reviewer #3: All comments have been addressed

2. Is the manuscript technically sound, and do the data support the conclusions?

Reviewer #1: No

Reviewer #3: Yes

3. Has the statistical analysis been performed appropriately and rigorously? 

Reviewer #1: No

Reviewer #3: Yes

4. Have the authors made all data underlying the findings in their manuscript fully available?

Reviewer #1: Yes

Reviewer #3: Yes

5. Is the manuscript presented in an intelligible fashion and written in standard English?

Reviewer #1: Yes

Reviewer #3: Yes

6. Review Comments to the Author

Reviewer #1: I apologize for being in the second revision for the manuscript after it was reviewed before but I have comments

As stated by the authors the number enrolled in the study is statistically low to generate a conclusion that can be accepted for practical use , I would recommend a multi centre collection of cases specially it is a retrospective study and can easily get higher number than selected

Also some other important points should be highlighted

1- From symptom onset to CXR is seen shorter in mild group compared to severe group despite that it is not statistically significant but as we know covid pneumonia is a progressive disease and if the patient is presenting late , his CXR might show more shadows, which might reflect late presentation in the severe group , so I can not judge that patients with mild CXR shadows are expected to have milder form of the disease, as it will progress in the few coming days , it was not stated in the methods patients who might come twice to the hospital with worsening symptoms and condition are excluded or not, if not which CXR was selected

2- As many cases of covid pneumonia are are afebrile throughout their illness , I feel fever in the table is of no value , and as oxygen saturation and respiratory rate are important vital signs in assessment of respiratory affection in covid pneumonia and it is easily assessed in all hospitals at presentation, I would recommend adding these parameters in the table beside or instead of fever

3- it was not mentioned in the methods who assessed the CXR shadows, and if they were aware about the patient clinical status and if there is any bias in their interpretation of CXR reading , is there any interobserver variability

Thanks

Reviewer #3: I think this is a good work. the authors have properly answered to all my questions, I have no further comments

7. PLOS authors have the option to publish the peer review history of their article (what does this mean?). If published, this will include your full peer review and any attached files.

Reviewer #1: **Yes: **Dr Hassan Aref Shabana

Reviewer #3: No

---

## [Author Response · Author response to Decision Letter 1]

27 Oct 2021

Reviewer #1, comment: As stated by the authors the number enrolled in the study is statistically low to generate a conclusion that can be accepted for practical use , I would recommend a multi centre collection of cases specially it is a retrospective study and can easily get higher number than selected

Response to reviewer #1, comment: in full agreement with the reviewer, we’ve already emphasized that small sample size is a limitation of our contribution. However, it is important to underline that all subjects were fully characterized, and consecutively collected in order to reduce selection bias. Of note, we’ve chosen not to extend the study population to the so called “second and third pandemic wave” because several treatments have been put in place (i.e. steroid treatment) which could have influenced the outcome. Indeed we believe that “first pandemic wave” represents a unique population which is difficult to merge with the subsequent pandemics. We agree with the reviewer that multicentric studies could lead to more significant results, however this topic will be addressed together with other centers in future studies. We added the monocentricity of this study to our limitations in the Discussion section (page 7, line 114-115).

Reviewer #1, comment 1: From symptom onset to CXR is seen shorter in mild group compared to severe group despite that it is not statistically significant but as we know covid pneumonia is a progressive disease and if the patient is presenting late , his CXR might show more shadows, which might reflect late presentation in the severe group , so I can not judge that patients with mild CXR shadows are expected to have milder form of the disease, as it will progress in the few coming days , it was not stated in the methods patients who might come twice to the hospital with worsening symptoms and condition are excluded or not, if not which CXR was selected 

Response to reviewer #1, comment 1: We agree with the reviewer on this point, actually even if there was no statistical difference in the two groups, we searched for multiple access to the ER. Of note, all patients have been hospitalized after the first ER admission. Following the reviewer’s comment we clarified this point in the Materials and Methods (page 4 lines 51-52) and Results sections (page 4 lines 69-70).

Reviewer #1, comment 2: As many cases of covid pneumonia are are afebrile throughout their illness , I feel fever in the table is of no value , and as oxygen saturation and respiratory rate are important vital signs in assessment of respiratory affection in covid pneumonia and it is easily assessed in all hospitals at presentation, I would recommend adding these parameters in the table beside or instead of fever

Response to reviewer #1, comment 2: We apologize, but the aim of our study was to replicate Toussie’s study on Radiology, so we kept the same parameters in our evaluation. We agree with the reviewer to add an evaluation of the gas exchanges, we used the P/F ratio, avoiding the bias of peripheral oxygen saturation due to oxygen supplementation. Confirming previous literature, P/F ratio was significantly lower (p=0.03) in HRRS patients. We then included this result in Table 1, in Results (page 6 lines 89-91) and Discussion section (page 7 lines 103-104). Unfortunately, respiratory rate was not routinely mentioned in medical records, and over 40% of the patients presented only semantic formulas like eupneic or tachipneic. Due to this lack of data we preferred not to include this parameter in our study.

Reviewer #1, comment 3: it was not mentioned in the methods who assessed the CXR shadows, and if they were aware about the patient clinical status and if there is any bias in their interpretation of CXR reading , is there any interobserver variability

Response to reviewer #1, comment 3: We thank the reviewer for this comment which is helpful to improve our contribution.; we therefore amended Materials and Methods (page 4 lines 54-56 and lines 60-61) and Results section (page 4 lines 71-73) including the required information.

Reviewer #3: I think this is a good work. the authors have properly answered to all my questions, I have no further comments

Response to reviewer #3: We thank the reviewer for the kind comment.

---

## [Decision Letter · Decision Letter 2]

7 Feb 2022

Brief communication: chest radiography score in young COVID-19 patients: does one size fit all?

PONE-D-21-07724R2

Dear Dr. Balestro,

We’re pleased to inform you that your manuscript has been judged scientifically suitable for publication and will be formally accepted for publication once it meets all outstanding technical requirements.

Kind regards,

Eman Sobh, M.D.

Academic Editor

PLOS ONE

Additional Editor Comments (optional):

Reviewers' comments:

Reviewer's Responses to Questions

**Comments to the Author**

1. If the authors have adequately addressed your comments raised in a previous round of review and you feel that this manuscript is now acceptable for publication, you may indicate that here to bypass the “Comments to the Author” section, enter your conflict of interest statement in the “Confidential to Editor” section, and submit your "Accept" recommendation.

Reviewer #1: All comments have been addressed

Reviewer #3: All comments have been addressed

2. Is the manuscript technically sound, and do the data support the conclusions?

Reviewer #1: Yes

Reviewer #3: Yes

3. Has the statistical analysis been performed appropriately and rigorously? 

Reviewer #1: Yes

Reviewer #3: Yes

4. Have the authors made all data underlying the findings in their manuscript fully available?

Reviewer #1: Yes

Reviewer #3: Yes

5. Is the manuscript presented in an intelligible fashion and written in standard English?

Reviewer #1: Yes

Reviewer #3: Yes

6. Review Comments to the Author

Reviewer #1: (No Response)

Reviewer #3: (No Response)

7. PLOS authors have the option to publish the peer review history of their article (what does this mean?). If published, this will include your full peer review and any attached files.

Reviewer #1: No

Reviewer #3: No

---

## [Editor Report · Acceptance letter]

14 Feb 2022

PONE-D-21-07724R2 

Brief communication: chest radiography score in young COVID-19 patients: does one size fit all? 

Dear Dr. Balestro:

I'm pleased to inform you that your manuscript has been deemed suitable for publication in PLOS ONE. Congratulations! Your manuscript is now with our production department. 

Kind regards, 

on behalf of

Dr. Eman Sobh 

Academic Editor

PLOS ONE